# Improving Adversarial Robustness Requires Revisiting Misclassified Examples

**Yisen Wang[1]**[*], **Difan Zou[2]**[*], **Jinfeng Yi[3]**, **James Bailey[4]**, **Xingjun Ma[4]**[†], **Quanquan Gu[2]**[†]
[1]Shanghai Jiao Tong University   [2]University of California, Los Angles
[3]JD.com   [4]The University of Melbourne
eewangyisen@gmail.com,   {knowzou, qgu}@cs.ucla.edu,
yijinfeng@jd.com,   {baileyj, xingjun.ma}@unimelb.edu.au

## Abstract

Deep neural networks (DNNs) are vulnerable to adversarial examples crafted by imperceptible perturbations. A range of defense techniques have been proposed to improve DNN robustness to adversarial examples, among which adversarial training has been demonstrated to be the most effective. Adversarial training is often formulated as a min-max optimization problem, with the inner maximization for generating adversarial examples. However, there exists a simple, yet easily overlooked fact that adversarial examples are only defined on correctly classified (natural) examples, but inevitably, some (natural) examples will be misclassified during training. In this paper, we investigate the distinctive influence of misclassified and correctly classified examples on the final robustness of adversarial training. Specifically, we find that misclassified examples indeed have a significant impact on the final robustness. More surprisingly, we find that different maximization techniques on misclassified examples may have a negligible influence on the final robustness, while different minimization techniques are crucial. Motivated by the above discovery, we propose a new defense algorithm called *Misclassification Aware adveRsarial Training* (MART), which explicitly differentiates the misclassified and correctly classified examples during the training. We also propose a semi-supervised extension of MART, which can leverage the unlabeled data to further improve the robustness. Experimental results show that MART and its variant could significantly improve the state-of-the-art adversarial robustness.

## 1 Introduction

Despite their great success in applications such as computer vision (He et al., 2016), speech recognition (Wang et al., 2017) and natural language processing (Devlin et al., 2018; Zeng et al., 2019), deep neural networks (DNNs) are extremely vulnerable to adversarial examples crafted by adding small adversarial perturbations to natural examples (Szegedy et al., 2013; Goodfellow et al., 2015; Wu et al., 2020). Given a DNN classifier $h_{\boldsymbol{\theta}}$ with parameter $\boldsymbol{\theta}$ and a correctly classified natural example $\mathbf{x}$ with class label $y$ ($h_{\boldsymbol{\theta}}(\mathbf{x}) = y$), an adversarial example $\mathbf{x}'$ can be generated by perturbing $\mathbf{x}$ such that $h_{\boldsymbol{\theta}}(\mathbf{x}') \neq y$, *i.e.*, the natural example is correctly classified before perturbation but misclassified after perturbation. The perturbation required for misclassification is often small and bounded by an $L_p$-norm $\|\mathbf{x}' - \mathbf{x}\|_p \leq \epsilon$, which keeps $\mathbf{x}'$ within the $\epsilon$-ball centered at $\mathbf{x}$, so that it is visually the "same" for human observers. This vulnerability of DNNs raises serious security concerns about their practicability in security critical applications (Chen et al., 2015; Kurakin et al., 2016; Jiang et al., 2019; Finlayson et al., 2019; Ma et al., 2019).

Compared with pre/post-processing methods such as feature squeezing (Xu et al., 2017), input denoising (Guo et al., 2018; Liao et al., 2018; Samangouei et al., 2018; Bai et al., 2019) and adversarial detection (Feinman et al., 2017; Ma et al., 2018; Lee et al., 2018), several defense techniques have been proposed to train DNNs that are inherently robust to adversarial examples including defensive distillation (Papernot et al., 2016), gradient regularization (Gu & Rigazio, 2014; Papernot et al., 2017; Ross & Doshi-Velez, 2018; Tramèr et al., 2018), model compression (Das et al., 2018; Liu et al., 2018) and activation pruning (Dhillon et al., 2018; Rakin et al., 2018), among which

---

[*]Equal contribution. [†]Corresponding author.

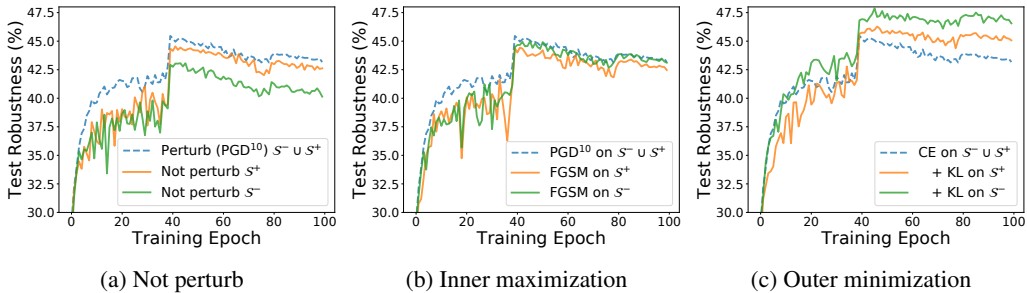

(a) Not perturb        (b) Inner maximization        (c) Outer minimization

Figure 1: The distinctive influence of misclassified examples ($\mathcal{S}^-$) versus correctly classified ones ($\mathcal{S}^+$) on the robustness of adversarial training. We test the white-box robustness of different strategies on either subset of examples: (a) using them directly for training ("not perturb"); (b) using weak attack (FGSM) in the inner maximization; and (c) using "regularized CE" loss in the outer minimization.

adversarial training has been demonstrated to be the most effective (Athalye et al., 2018). Adversarial training can be regarded as a data augmentation technique that trains DNNs on adversarial examples, and can be viewed as solving the following min-max optimization problem (Madry et al., 2018):

$$\min_{\boldsymbol{\theta}} \frac{1}{n} \sum_{i=1}^{n} \max_{\|\mathbf{x}_i' - \mathbf{x}_i\|_p \leq \epsilon} \ell(h_{\boldsymbol{\theta}}(\mathbf{x}_i'), y_i), \tag{1}$$

where $n$ is the number of training examples and $\ell(\cdot)$ is the classification loss, such as the commonly used cross-entropy (CE) loss. The inner maximization generates adversarial examples that can be used by the outer minimization to train robust DNNs. Recently, adversarial training with adversarial examples generated by Projected Gradient Descent (PGD) (Madry et al., 2018) has been demonstrated to be the only method that can train moderately robust DNNs without being fully attacked (Athalye et al., 2018). However, there is still a significant gap between adversarial robustness (test accuracy on adversarial examples) and natural accuracy (test accuracy on natural examples), even for simple image datasets like CIFAR-10 (Krizhevsky & Hinton, 2009).

Compared with natural training (on natural examples), training adversarially robust DNNs is particularly difficult (Madry et al., 2018). Nakkiran (2019) showed that a model requires more capacity to be robust (*i.e.*, simple models can have high natural accuracy but are less likely to be robust). In addition, the sample complexity of adversarial training can be significantly higher than that of natural training, that is, training robust DNNs tends to require more data either labeled (Schmidt et al., 2018) or unlabeled ones (Uesato et al., 2019; Carmon et al., 2019; Najafi et al., 2019; Zhai et al., 2019). Moreover, Tsipras et al. (2019); Zhang et al. (2019) demonstrated that adversarial robustness may be inherently at odds with natural accuracy. Parallel to these studies, in this paper, we provide some new insights on the adversarial examples used for adversarial training.

Recall that the formal definition of an adversarial example is conditioned on it being correctly classified[*] (Carlini et al., 2019). From this perspective, adversarial examples generated from misclassified examples are "undefined". Most adversarial training variants neglect this distinction, where all training examples are treated equally in both the maximization and the minimization processes, regardless of whether or not they are correctly classified. The only exception we are aware of is Ding et al. (2018), which proposes to use maximal margin optimization for correctly classified examples. Yet they did not pay sufficient attention to misclassified examples. A deeper understanding about the influence of misclassified and correctly classified examples on the robustness is still missing in the literature. Therefore, we raise the following questions:

> *Are the adversarial examples generated from i) misclassified and ii) correctly classified examples, equally important for adversarial robustness?* **If not**, *how can one make better use of the difference to improve robustness?*

In this paper, we investigate this intriguing, yet thus far overlooked aspect of adversarial training, and find that misclassified and correctly classified examples exhibit a distinctive influence on the final robustness. To illustrate this phenomenon, we conduct a proof-of-concept experiment on CIFAR-10 in a white-box setting with $L_\infty$ maximum perturbation $\epsilon = 8/255$. We first train an 8-layer

---

[*]In this paper, both "correctly classified" and "misclassified" refer to predictions on *natural* training examples.

Convolutional Neural Network (CNN) using standard adversarial training with 10-step PGD (PGD[10]) and step size $\epsilon/4$, then use this network (87% training accuracy) to select two subsets of natural training examples to investigate: 1) a subset of misclassified examples $\mathcal{S}^-$ (13% of training data), and 2) a subset of correctly classified examples $\mathcal{S}^+$ (also 13% of training data, $|\mathcal{S}^+| = |\mathcal{S}^-|$). Using these two subsets, we explore different ways to re-train the same network, and evaluate its robustness against white-box PGD[20] (step size $\epsilon/10$) attacks on the test dataset.

In Figure 1(a), we find that misclassified examples have a significant impact on the final robustness. Compared with standard adversarial training (dashed blue line), the final robustness drops drastically, if examples in subset $\mathcal{S}^-$ are not perturbed (solid green line) during adversarial training (other examples are still perturbed by PGD[10]). In contrast, the same operation on subset $\mathcal{S}^+$ only slightly affects the final robustness (solid orange line). Previous work has found that removing a small proportion of training examples does not reduce the robustness (Ding et al., 2019), which seems to be true for correctly classified examples, but is apparently not true for misclassified examples.

To further understand the distinctive influence of misclassified and correctly classified examples, we test different techniques on them within either the maximization or the minimization process of adversarial training. Firstly, we apply different maximization techniques while keeping the minimization loss CE unchanged. As shown in Figure 1(b), the final robustness is barely affected when we use a weak attack (*e.g.*, Fast Gradient Sign Method (FGSM) (Goodfellow et al., 2015)) to perturb misclassified examples $\mathcal{S}^-$ (all other training examples are still perturbed by PGD[10]). This suggests that different maximization techniques on misclassified examples $\mathcal{S}^-$ may have a negligible influence on the final robustness, provided that the inner maximization problem is solved to a moderate precision (Wang et al., 2019). However, for subset $\mathcal{S}^+$, a weak attack for the maximization tends to degenerate the robustness. Secondly, we test different minimization techniques with the inner maximization still solved by PGD[10]. Interestingly, we find that different minimization techniques on misclassified examples make a significant difference to the final robustness. As shown in Figure 1(c), compared with standard adversarial training (dashed blue line) with the CE loss, the final robustness is significantly improved when the outer minimization on misclassified examples is "regularized" (solid green line) by an additional term (a KL-divergence term that was used previously in Zheng et al. (2016); Zhang et al. (2019)). The same regularization applied to correctly classified examples also helps the final robustness (solid orange line), though not as significantly as for misclassified examples.

Motivated by the above observations, we reformulate the adversarial risk to incorporate an explicit differentiation of misclassified examples in a form of regularization. We then propose a new defense algorithm to achieve this in a dynamic way during adversarial training. Our main contributions are:

- We investigate the distinctive influence of misclassified and correctly classified examples on the final robustness of adversarial training. We find that the manipulation on misclassified examples has more impact on the final robustness, and the minimization techniques are more crucial than maximization ones under the min-max optimization framework.

- We propose a regularized adversarial risk which incorporates an explicit differentiation of misclassified examples as a regularizer. Based on that, we further propose a new defense algorithm, called *Misclassification Aware adveRsarial Training* (MART).

- Experimentally, we show that adversarial robustness can be significantly improved over the state-of-the-art, by a specific focus on misclassified examples. It also helps improve recently proposed adversarial training with unlabeled data.

## 2 MISCLASSIFICATION AWARE ADVERSARIAL RISK

In this section, we propose a regularized adversarial risk that incorporates an explicit differentiation of misclassified examples.

### 2.1 PRELIMINARIES

We first define some notations. We use lower case and lower case bold face to denote scalars and vectors, respectively. We use upper case calligraphic symbols to denote sets.

For a $K$-class ($K \geq 2$) classification problem, given a dataset $\{(\mathbf{x}_i, y_i)\}_{i=1,\ldots,n}$ with $\mathbf{x}_i \in \mathbb{R}^d$ as a natural example and $y_i \in \{1, \ldots, K\}$ as its associated label, a DNN classifier $h_{\boldsymbol{\theta}}$ with model

parameter $\boldsymbol{\theta}$ predicts the class of an input example $\mathbf{x}_i$:

$$h_{\boldsymbol{\theta}}(\mathbf{x}_i) = \arg\max_{k=1,\dots,K} \mathbf{p}_k(\mathbf{x}_i, \boldsymbol{\theta}), \quad \mathbf{p}_k(\mathbf{x}_i, \boldsymbol{\theta}) = \exp(\mathbf{z}_k(\mathbf{x}_i, \boldsymbol{\theta})) / \sum_{k'=1}^{K} \exp(\mathbf{z}_{k'}(\mathbf{x}_i, \boldsymbol{\theta})), \quad (2)$$

where $\mathbf{z}_k(\mathbf{x}_i, \boldsymbol{\theta})$ is the logits output of the network with respect to class $k$, and $\mathbf{p}_k(\mathbf{x}_i, \boldsymbol{\theta})$ is the probability (softmax on logits) of $\mathbf{x}_i$ belonging to class $k$.

The adversarial risk (Madry et al., 2018) on dataset $\{(\mathbf{x}_i, y_i)\}_{i=1,\dots,n}$ and classifier $h_{\boldsymbol{\theta}}$ can be defined with respect to the 0-1 loss (Zhang et al., 2019) as:

$$\mathcal{R}(h_{\boldsymbol{\theta}}) = \frac{1}{n} \sum_{i=1}^{n} \max_{\mathbf{x}_i' \in \mathcal{B}_\epsilon(\mathbf{x}_i)} \mathbb{1}\big(h_{\boldsymbol{\theta}}(\mathbf{x}_i') \neq y_i\big), \quad (3)$$

where $\mathbb{1}(\cdot)$ is the indicator function and $\mathcal{B}_\epsilon(\mathbf{x}_i) = \{\mathbf{x} : \|\mathbf{x} - \mathbf{x}_i\|_p \leq \epsilon\}$ denotes the $L_p$-norm ball centered at $\mathbf{x}_i$ with radius $\epsilon$. We will focus on the $L_\infty$-ball in this paper.

## 2.2 MISCLASSIFICATION AWARE REGULARIZATION

Note that the adversarial risk in (3) is defined on adversarial examples within the $\epsilon$-ball of all natural examples, regardless of whether they are correctly classified ($h_{\boldsymbol{\theta}}(\mathbf{x}_i) = y_i$) or misclassified ($h_{\boldsymbol{\theta}}(\mathbf{x}_i) \neq y_i$) by the current model $h_{\boldsymbol{\theta}}$. To differentiate, we reformulate the adversarial risk based on the prediction of the current network $h_{\boldsymbol{\theta}}$. Specifically, natural training examples can be divided into two subsets with respect to $h_{\boldsymbol{\theta}}$, with one subset of correctly classified examples ($\mathcal{S}_{h_{\boldsymbol{\theta}}}^+$) and one subset of misclassified examples ($\mathcal{S}_{h_{\boldsymbol{\theta}}}^-$):

$$\mathcal{S}_{h_{\boldsymbol{\theta}}}^+ = \{i : i \in [n], h_{\boldsymbol{\theta}}(\mathbf{x}_i) = y_i\} \quad \text{and} \quad \mathcal{S}_{h_{\boldsymbol{\theta}}}^- = \{i : i \in [n], h_{\boldsymbol{\theta}}(\mathbf{x}_i) \neq y_i\}.$$

Then we are going to define adversarial risk separately for correctly classified and misclassified examples. As we observed in Figure 1(c), regularization on misclassified examples can significantly improve robustness. Therefore, for misclassified examples, we formulate the adversarial risk as:

$$\mathcal{R}^-(h_{\boldsymbol{\theta}}, \mathbf{x}_i) := \mathbb{1}(h_{\boldsymbol{\theta}}(\hat{\mathbf{x}}_i') \neq y_i) + \mathbb{1}(h_{\boldsymbol{\theta}}(\mathbf{x}_i) \neq h_{\boldsymbol{\theta}}(\hat{\mathbf{x}}_i')), \quad (4)$$

where the adversarial example $\hat{\mathbf{x}}_i'$ is generated by solving

$$\hat{\mathbf{x}}_i' = \arg\max_{\mathbf{x}_i' \in \mathcal{B}_\epsilon(\mathbf{x}_i)} \mathbb{1}(h_{\boldsymbol{\theta}}(\mathbf{x}_i') \neq y_i). \quad (5)$$

We remark that the first and second terms on the R.H.S. of (4) correspond to the standard adversarial risk and the regularization term respectively. Moreover, we would like to clarify that the regularization term $\mathbb{1}(h_{\boldsymbol{\theta}}(\mathbf{x}_i) \neq h_{\boldsymbol{\theta}}(\hat{\mathbf{x}}_i'))$ aims to encourage the output of neural network to be stable against misclassified adversarial examples. For misclassified examples, direct minimization of the standard adversarial risk may be too hard, as themselves cannot be classified correctly, even without any perturbations. A similar idea has been used in stability training on all training examples (Zheng et al., 2016; Kannan et al., 2018; Zhang et al., 2019).

Then we consider correctly classified examples. As can be observed in Figure 1(c), regularization on correctly classified examples cannot provide as significant improvement as achieved by that on misclassified ones. Moreover, in this case it can be found that $\mathbb{1}(h_{\boldsymbol{\theta}}(\mathbf{x}_i) \neq h_{\boldsymbol{\theta}}(\hat{\mathbf{x}}_i')) = \mathbb{1}(h_{\boldsymbol{\theta}}(\hat{\mathbf{x}}_i') \neq y_i)$ since we have $h_{\boldsymbol{\theta}}(\mathbf{x}_i) = y_i$, which implies that the regularizer has exactly same form as the adversarial risk. Therefore, for correctly classified example, we simply use the *standard* adversarial risk, *i.e.*,

$$\mathcal{R}^+(h_{\boldsymbol{\theta}}, \mathbf{x}_i) := \max_{\mathbf{x}_i' \in \mathcal{B}_\epsilon(\mathbf{x}_i)} \mathbb{1}(h_{\boldsymbol{\theta}}(\mathbf{x}_i') \neq y_i) = \mathbb{1}(h_{\boldsymbol{\theta}}(\hat{\mathbf{x}}_i') \neq y_i), \quad (6)$$

where the adversarial example $\hat{\mathbf{x}}_i'$ is defined in (5).

Finally, combining the proposed two adversarial risks for correctly classified examples and misclassified examples in an adversarial training framework, we can train a network that minimizes the following risk:

$$\min_{\boldsymbol{\theta}} \mathcal{R}_{\mathrm{misc}}(h_{\boldsymbol{\theta}}) := \frac{1}{n} \Big( \sum_{i \in \mathcal{S}_{h_{\boldsymbol{\theta}}}^+} \mathcal{R}^+(h_{\boldsymbol{\theta}}, \mathbf{x}_i) + \sum_{i \in \mathcal{S}_{h_{\boldsymbol{\theta}}}^-} \mathcal{R}^-(h_{\boldsymbol{\theta}}, \mathbf{x}_i) \Big)$$

$$= \frac{1}{n} \sum_{i=1}^{n} \big\{ \mathbb{1}(h_{\boldsymbol{\theta}}(\hat{\mathbf{x}}_i') \neq y_i) + \mathbb{1}(h_{\boldsymbol{\theta}}(\mathbf{x}_i) \neq h_{\boldsymbol{\theta}}(\hat{\mathbf{x}}_i')) \cdot \mathbb{1}(h_{\boldsymbol{\theta}}(\mathbf{x}_i) \neq y_i) \big\}, \quad (7)$$

where $\hat{\mathbf{x}}_i'$ is defined in (5) and the second equality follows from the definition of $\mathcal{S}_{h_{\boldsymbol{\theta}}}^+$ and $\mathcal{S}_{h_{\boldsymbol{\theta}}}^-$. The new risk defined above is a regularized adversarial risk with regularization term $1/n \sum_{i=1}^{n} \mathbb{1}(h_{\boldsymbol{\theta}}(\mathbf{x}_i) \neq h_{\boldsymbol{\theta}}(\hat{\mathbf{x}}_i')) \cdot \mathbb{1}(h_{\boldsymbol{\theta}}(\mathbf{x}_i) \neq y_i)$, which we call the *misclassification aware regularization*.

## 3 PROPOSED DEFENSE: MISCLASSIFICATION AWARE ADVERSARIAL TRAINING (MART)

In the previous section, we derived the misclassification aware adversarial risk based on 0-1 loss. However, optimization over 0-1 loss is intractable in practice. We next propose a Misclassification Aware adveRsarial Training (MART) algorithm, by replacing the 0-1 losses with proper surrogate loss functions which are both physical meaningful and computationally tractable. Following that, we further analyze the difference of MART to existing work, and propose a semi-supervised extension.

### 3.1 THE PROPOSED DEFENSE ALGORITHM

**Surrogate Loss for Outer Minimization.** As presented in (7), the minimization consists of three indicator functions: (1) $\mathbb{1}(h_{\boldsymbol{\theta}}(\hat{\mathbf{x}}_i') \neq y_i)$; (2) $\mathbb{1}(h_{\boldsymbol{\theta}}(\mathbf{x}_i) \neq h_{\boldsymbol{\theta}}(\hat{\mathbf{x}}_i'))$; and (3) $\mathbb{1}(h_{\boldsymbol{\theta}}(\mathbf{x}_i) \neq y_i)$.

For the first indicator function $\mathbb{1}(h_{\boldsymbol{\theta}}(\hat{\mathbf{x}}_i') \neq y_i)$, we propose to use a boosted cross entropy (BCE) loss as the surrogate loss, instead of the commonly used CE loss in (Madry et al., 2018; Wang et al., 2019). This is largely because classifying adversarial examples requires a stronger classifier than natural examples, as the presence of adversarial examples makes the classification decision boundary become more complicated. This is pointed out by (Madry et al., 2018), where they increase the model capacity for a stronger classifier. The benefit of using BCE compared to CE will shortly be presented in the experiment section. The proposed BCE loss is defined as:

$$\text{BCE}\big(\mathbf{p}(\hat{\mathbf{x}}_i', \boldsymbol{\theta}), y_i\big) = -\log\big(\mathbf{p}_{y_i}(\hat{\mathbf{x}}_i', \boldsymbol{\theta})\big) - \log\big(1 - \max_{k \neq y_i} \mathbf{p}_k(\hat{\mathbf{x}}_i', \boldsymbol{\theta})\big), \tag{8}$$

where $\mathbf{p}_k(\hat{\mathbf{x}}_i', \boldsymbol{\theta})$ is the probability output defined in (2), the first term $-\log\big(\mathbf{p}_{y_i}(\hat{\mathbf{x}}_i', \boldsymbol{\theta})\big)$ is the commonly used CE loss, denoted $\text{CE}(\mathbf{p}(\hat{\mathbf{x}}_i', \boldsymbol{\theta}), y_i)$, and the second term $-\log\big(1 - \max_{k \neq y_i} \mathbf{p}_k(\hat{\mathbf{x}}_i', \boldsymbol{\theta})\big)$ is a margin term used to improve the decision margin of the classifier. A similar idea has been used for improving adversarial strength by Carlini & Wagner (2017). Note that BCE is just a simple boost that works well in our experiments and other boosted losses could also work here.

For the second indicator function $\mathbb{1}(h_{\boldsymbol{\theta}}(\mathbf{x}_i) \neq h_{\boldsymbol{\theta}}(\hat{\mathbf{x}}_i'))$, we can use KL divergence as the surrogate loss function (Zhang et al., 2019; Zheng et al., 2016), since $h_{\boldsymbol{\theta}}(\mathbf{x}_i) \neq h_{\boldsymbol{\theta}}(\hat{\mathbf{x}}_i')$ implies that adversarial examples have different output distributions to that of natural examples. Thus, we have

$$\text{KL}\big(\mathbf{p}(\mathbf{x}_i, \boldsymbol{\theta}) \| \mathbf{p}(\hat{\mathbf{x}}_i', \boldsymbol{\theta})\big) = \sum_{k=1}^{K} \mathbf{p}_k(\mathbf{x}_i, \boldsymbol{\theta}) \log \frac{\mathbf{p}_k(\mathbf{x}_i, \boldsymbol{\theta})}{\mathbf{p}_k(\hat{\mathbf{x}}_i', \boldsymbol{\theta})}. \tag{9}$$

The third indicator function $\mathbb{1}(h_{\boldsymbol{\theta}}(\mathbf{x}_i) \neq y_i)$ is a condition that emphasizes learning on misclassified examples. However, the condition cannot be directly optimized if we conduct a hard decision during the training process (Ding et al. (2019) uses hard decision and does not optimize the condition). Instead, we propose to use a soft decision scheme by replacing $\mathbb{1}(h_{\boldsymbol{\theta}}(\mathbf{x}_i) \neq y_i)$ with the output probability $1 - \mathbf{p}_{y_i}(\mathbf{x}_i, \boldsymbol{\theta})$. This will be large for misclassified examples and small for correctly classified examples.

**Surrogate Loss for Inner Maximization.** The goal of inner maximization is to generate adversarial example $\hat{\mathbf{x}}_i'$ for natural example $\mathbf{x}_i$ by solving (5). Therefore, we aim to find a surrogate loss function for the indicator function $\mathbb{1}(h_{\boldsymbol{\theta}}(\mathbf{x}_i') \neq y_i)$. Here, we leverage the commonly used CE loss as the surrogate loss and find the adversarial example $\hat{\mathbf{x}}_i'$ [†] as follows:

$$\hat{\mathbf{x}}_i' = \underset{\mathbf{x}_i' \in \mathcal{B}_\epsilon(\mathbf{x}_i)}{\arg\max} \text{CE}\big(\mathbf{p}(\mathbf{x}_i', \boldsymbol{\theta}), y_i\big). \tag{10}$$

Following our findings in Figure 1(b) that a strong attack can help robustness (though is negligible for misclassified examples), we propose to use the (strong) PGD attack to maximize the CE loss for both the correctly classified and misclassified examples, the same as standard adversarial training. Note that other surrogate loss functions, such as those exploited in (Athalye et al., 2018; Carlini & Wagner, 2017) for adversarial attack, could also be used here.

---

[†]We slightly abuse the notation of $\hat{\mathbf{x}}_i'$ as the maximizer of the surrogate loss rather than the 0-1 loss in (5).

**The Overall Objective.** Based on the surrogate loss functions, we can state the final objective function for our proposed *Misclassification Aware adveRsarial Training* (MART) defense:

$$\mathcal{L}^{\mathrm{MART}}(\boldsymbol{\theta}) = \frac{1}{n} \sum_{i=1}^{n} \ell(\mathbf{x}_i, y_i, \boldsymbol{\theta}), \tag{11}$$

where $\ell(\mathbf{x}_i, y_i, \boldsymbol{\theta})$ is defined as

$$\ell(\mathbf{x}_i, y_i, \boldsymbol{\theta}) := \mathrm{BCE}\big(\mathbf{p}(\hat{\mathbf{x}}_i', \boldsymbol{\theta}), y_i\big) + \lambda \cdot \mathrm{KL}\big(\mathbf{p}(\mathbf{x}_i, \boldsymbol{\theta}) \| \mathbf{p}(\hat{\mathbf{x}}_i', \boldsymbol{\theta})\big) \cdot \big(1 - \mathbf{p}_{y_i}(\mathbf{x}_i, \boldsymbol{\theta})\big).$$

Here the adversarial example $\hat{\mathbf{x}}_i'$ is generated by (10), and $\lambda$ is a tunable scaling parameter that balances the two parts of the final loss, and is fixed for all training examples. The complete training procedure of MART is described in Appendix A.

## 3.2 RELATION TO EXISTING WORK

In this section, we briefly discuss the difference between our MART and existing defense methods including standard adversarial training (*Standard*) (Madry et al., 2018), logit pairing methods (Kannan et al., 2018), max-margin adversarial training (MMA) (Ding et al., 2018) and TRADES (Zhang et al., 2019), as presented in Table 1.

Table 1: Loss function comparison with existing work. The adversarial example $\hat{\mathbf{x}}'$ is generated by (10) for all defense methods except TRADES and MMA. The adversarial example in TRADES is generated by maximizing its regularization term (KL-divergence), and the adversarial example in MMA is generated by solving (10) with different perturbation limit (*i.e.*, $\epsilon$).

| Defense Method | Loss Function |
|---|---|
| *Standard* | $\mathrm{CE}(\mathbf{p}(\hat{\mathbf{x}}', \boldsymbol{\theta}), y)$ |
| ALP | $\mathrm{CE}(\mathbf{p}(\hat{\mathbf{x}}', \boldsymbol{\theta}), y) + \lambda \cdot \|\mathbf{p}(\hat{\mathbf{x}}', \boldsymbol{\theta}) - \mathbf{p}(\mathbf{x}, \boldsymbol{\theta})\|_2^2$ |
| CLP | $\mathrm{CE}(\mathbf{p}(\mathbf{x}, \boldsymbol{\theta}), y) + \lambda \cdot \|\mathbf{p}(\hat{\mathbf{x}}', \boldsymbol{\theta}) - \mathbf{p}(\mathbf{x}, \boldsymbol{\theta})\|_2^2$ |
| TRADES | $\mathrm{CE}(\mathbf{p}(\mathbf{x}, \boldsymbol{\theta}), y) + \lambda \cdot \mathrm{KL}\big(\mathbf{p}(\mathbf{x}, \boldsymbol{\theta}) \| \mathbf{p}(\hat{\mathbf{x}}', \boldsymbol{\theta})\big)$ |
| MMA | $\mathrm{CE}(\mathbf{p}(\hat{\mathbf{x}}', \boldsymbol{\theta}), y) \cdot \mathbb{1}(h_{\boldsymbol{\theta}}(\mathbf{x}) = y) + \mathrm{CE}(\mathbf{p}(\mathbf{x}, \boldsymbol{\theta}), y) \cdot \mathbb{1}(h_{\boldsymbol{\theta}}(\mathbf{x}) \neq y)$ |
| **MART** | $\mathrm{BCE}(\mathbf{p}(\hat{\mathbf{x}}', \boldsymbol{\theta}), y) + \lambda \cdot \mathrm{KL}(\mathbf{p}(\mathbf{x}, \boldsymbol{\theta}) \| \mathbf{p}(\hat{\mathbf{x}}', \boldsymbol{\theta})) \cdot (1 - \mathbf{p}_y(\mathbf{x}, \boldsymbol{\theta}))$ |

Specifically, the *Standard* algorithm was designed to minimize the standard adversarial loss, *i.e.*, cross-entropy loss on adversarial examples. Logit pairing methods, consisting of adversarial logit pairing (ALP) and clean logit pairing (CLP), introduce a regularization term enclosing both natural examples and their adversarial counterparts. The objective function of TRADES is also a linear combination of natural loss and regularization terms on the output probabilities corresponding to natural examples and their adversarial counterparts using KL divergence. However, none of these algorithms differentiates the misclassified examples and correctly classified examples.

The most relevant work is MMA, which proposes to use maximal margin optimization for correctly classified examples while keeping the optimization on misclassified examples unchanged. Specifically, for correctly classified examples, MMA adopts cross-entropy loss on adversarial examples, which are generated by solving (10) with example-dependent perturbation limit. For misclassified examples, MMA directly applies cross-entropy loss on natural examples. We emphasize that our MART is different from MMA in the following aspects: (1) MMA performs hard decision to identify misclassified examples from training data, while MART uses soft decision scheme on training data based on the corresponding output probabilities ($\mathbf{p}(\hat{\mathbf{x}}', \boldsymbol{\theta})$), which can be jointly learned during the training process; (2) for correctly classified examples, MMA adopts cross-entropy loss on adversarial examples with different perturbation limits, while MART utilizes the proposed BCE loss on the adverarial examples with the same perturbation limit; (3) for misclassified examples, MMA adopts cross-entropy loss on natural examples, while MART adopts a regularized adversarial loss involving both adversarial and natural examples. Because of these differences, we later will show that MART outperforms MMA in the experiments.

## 3.3 SEMI-SUPERVISED EXTENSION WITH UNLABELED DATA

Recent work has shown that semi-supervised learning with additional unlabeled data can improve the adversarial robustness (Uesato et al., 2019; Carmon et al., 2019; Najafi et al., 2019; Zhai et al., 2019). Specifically, the training loss function applied in these semi-supervised learning methods is typically

defined as a weighted sum of the supervised loss (loss on the labeled data) and the unsupervised one (loss on the unlabeled data), *i.e.*,

$$\mathcal{L}(\boldsymbol{\theta}) = \mathcal{L}_{\text{sup}}(\boldsymbol{\theta}) + \gamma \cdot \mathcal{L}_{\text{unsup}}(\boldsymbol{\theta}),$$

where $\gamma > 0$ is the weight of unsupervised loss. As pointed out in Uesato et al. (2019), there are multiple choices of the unsupervised loss function $\mathcal{L}_{\text{unsup}}(\boldsymbol{\theta})$, leading to different defense methods, among which the most effective defense method is UAT++. In particular, UAT++ first trains a natural model on labeled data, and then use this model to generate pseudo labels for unlabeled data. Moreover, given a training data $(\mathbf{x}, y)$ (which can be either labeled or unlabeled data), the supervised and unsupervised loss functions adopted in UAT++ are defined as[‡]

$$\ell_{\text{sup}}^{\text{UAT++}}(\mathbf{x}, y; \boldsymbol{\theta}) = \ell_{\text{unsup}}^{\text{UAT++}}(\mathbf{x}, y; \boldsymbol{\theta}) = \max_{\mathbf{x}' \in \mathcal{B}_\epsilon} \text{CE}(\mathbf{p}(\mathbf{x}', \boldsymbol{\theta}), y) + \lambda \cdot \max_{\mathbf{x}' \in \mathcal{B}_\epsilon} \text{KL}(\mathbf{p}(\mathbf{x}, \boldsymbol{\theta}) || \mathbf{p}(\mathbf{x}', \boldsymbol{\theta})),$$

where $\lambda$ is a tunable hyperparameter. A similar idea was also proposed in a concurrent work (Carmon et al., 2019), leading to another semi-supervised defense method called RST. The first stage of RST is also to generate pseudo labels for unlabeled data by training a natural model on the labeled data. Then in the second stage, RST applies TRADES loss to train the robust model based on both labeled and unlabeled data, *i.e.*, given a training data $(\mathbf{x}, y)$, the supervised and unsupervised loss functions adopted in RST is defined as

$$\ell_{\text{sup}}^{\text{RST}}(\mathbf{x}, y; \boldsymbol{\theta}) = \ell_{\text{unsup}}^{\text{RST}}(\mathbf{x}, y; \boldsymbol{\theta}) = \text{CE}(\mathbf{p}(\mathbf{x}, \boldsymbol{\theta}), y) + \lambda \cdot \max_{\mathbf{x}' \in \mathcal{B}_\epsilon} \text{KL}(\mathbf{p}(\mathbf{x}, \boldsymbol{\theta}) || \mathbf{p}(\mathbf{x}', \boldsymbol{\theta})).$$

As we pointed out in Figure 1(b) and the following experiment section, the maximization technique has a neglectable influence on the robustness. Therefore, the major difference between UAT++ and RST is the objective function for minimization. Considering MART is also an objective function, it thus could be easily combined with semi-supervised learning with unlabeled data. Following RST, we propose the following semi-supervised version of MART:

$$\mathcal{L}_{\text{semi}}^{\text{MART}}(\boldsymbol{\theta}) = \sum_{i \in \mathcal{S}_{\text{sup}}} \ell_{\text{sup}}^{\text{MART}}(\mathbf{x}_i, y_i; \boldsymbol{\theta}) + \gamma \cdot \sum_{i \in \mathcal{S}_{\text{unsup}}} \ell_{\text{unsup}}^{\text{MART}}(\mathbf{x}_i, y_i; \boldsymbol{\theta})$$

with supervised and unsupervised loss function defined as follows,

$$\ell_{\text{sup}}^{\text{MARTT}}(\mathbf{x}, y; \boldsymbol{\theta}) = \ell_{\text{unsup}}^{\text{MART}}(\mathbf{x}, y; \boldsymbol{\theta}) = \text{BCE}(\mathbf{p}(\hat{\mathbf{x}}', \boldsymbol{\theta}), y) + \lambda \cdot \text{KL}(\mathbf{p}(\mathbf{x}, \boldsymbol{\theta}) || \mathbf{p}(\hat{\mathbf{x}}', \boldsymbol{\theta})) \cdot (1 - \mathbf{p}_y(\mathbf{x}, \boldsymbol{\theta})),$$

where the adversarial example $\hat{\mathbf{x}}'$ is generated by solving (10), and $\mathcal{S}_{\text{sup}}$ and $\mathcal{S}_{\text{unsup}}$ denote the set of labeled data and unlabeled data respectively.

## 4    EXPERIMENTS

In this section, we first conduct a set of experiments to provide a comprehensive understanding of our proposed defense MART, and then evaluate its robustness on benchmark datasets in both white-box and black-box settings. Finally, we benchmark the state-of-the-art robustness and explore using unlabeled data for further improvement.

### 4.1    UNDERSTANDING THE PROPOSED MART

Here, we investigate MART from 4 different perspectives: (1) removing components of the MART loss function, (2) replacing components of the MART loss function, (3) misclassification aware loss on certain proportions of training data, and (4) sensitivity to regularization parameter $\lambda$.

**Experimental Setup.** We train ResNet-18 (He et al., 2016) with different variants of MART on CIFAR-10 (Krizhevsky & Hinton, 2009). All the models are trained using SGD with momentum 0.9, weight decay $2 \times 10^{-4}$ and an initial learning rate of 0.1, which is divided by 10 at the 75-th and 90-th epoch. All natural images are normalized into [0, 1], and simple data augmentations including 4-pixel padding with $32 \times 32$ random crop and random horizontal flip. The maximum perturbation $\epsilon = 8/255$ and parameter $\lambda = 6$. The training attack is $\text{PGD}^{10}$ with random start and step size $\epsilon/4$, while the test attack is $\text{PGD}^{20}$ with random start and step size $\epsilon/10$.

**Removing Components of MART.** Recalling the objective function of MART in (11), it has three terms in the loss function: BCE, KL and $1 - p$ [§]. As illustrated in Figure 2(a), removing $1 - p$ or KL

---

[‡]The adversarial example $\mathbf{x}'$ in KL term is reused the one in CE term for efficiency in Uesato et al. (2019)
[§]For simplicity, we use these abbreviations in Section 4.1

(a) Removing      (b) Replacing      (c) Training data (%)      (d) Regularization $\lambda$ ¶

Figure 2: The comprehensive ablation experiments of MART. In each plot, the dashed blue line represents the original MART method.

or both all leads to a significant robustness degradation. In particular, we found that the soft decision term $1 - p$ has a constant robustness improvement throughout the training process, while the KL term can help mitigate overfitting at a later stage of training (after 80 epochs). When the two terms are combined together, they boost the final robustness considerably without causing overfitting.

**Replacing Components of MART.** As we show in Figure 2(b), when the BCE component is either replaced by a CE term or redefined on natural examples ($\mathbf{x}_{nat}$), the final robustness decreases by a substantial amount. It suggests that learning with CE instead of our proposed BCE suffers from insufficient learning with lower robustness throughout the entire training process. On the other hand, learning with BCE on natural examples exhibits severe overfitting at the later stage (solid green line). We did not observe any benefit when replacing CE by KL in the inner maximization of adversarial min-max framework (solid red line), an observation that is consistent with Figure 1(b).

**Ablation on Training Data.** Here, we show the contribution of our proposed misclassification aware regularization (*e.g.*, the KL $\cdot (1 - p)$ term in (11)) to the final robustness with respect to the training data. Specifically, we gradually increase the proportion of training examples that are trained using the proposed misclassification aware regularization term, and display the corresponding robustness in Figure 2(c). The training examples using the proposed regularization are randomly selected, and the BCE term is still defined on all training (adversarial) examples. As can be observed, the robustness can be improved steadily when the proposed regularization is applied on more data. This verifies the benefit of the differentiation of correctly classified and misclassified examples.

**Sensitivity to Regularization Parameter $\lambda$.** We further investigate the parameter $\lambda$ in MART objective function defined in (11) which controls the strength of the regularization. We also test the regularization parameter $\lambda$ of TRADES (please refer to Table 1). We present the results in Figure 2(d)¶ for different $\lambda \in [1/2, 50]$. By explicitly differentiating the misclassified and correctly classified examples, MART achieves good stability and robustness across different choices of $\lambda$, which is also consistently better and more stable than that of TRADES.

### 4.2 ROBUSTNESS EVALUATION AND ANALYSIS

In this part, we evaluate the robustness of MART on both MNIST (LeCun et al., 1998) and CIFAR-10 (Krizhevsky & Hinton, 2009) datasets against various white-box and black-box attacks.

**Baselines.**‖ (1) *Standard* (Madry et al., 2018); (2) MMA (Ding et al., 2019); (3) *Dynamic* (Wang et al., 2019); and (4) TRADES (Zhang et al., 2019). We only compare with adversarial training variants, since they are the most effective defense to date (Athalye et al., 2018).

**Defense Settings.** For MNIST, all defense models are built on a 4-layer CNN and trained using SGD with momentum 0.9. The initial learning rate is 0.01 and divided by 10 at the 20-th and 40-th epoch. For CIFAR-10, we use SGD with momentum 0.9, weight decay $3.5 \times 10^{-3}$ and an initial learning rate of 0.01, which is divided by 10 at the 75-th and 90-th epoch. For the training attack, it is also the PGD[10] with random start and step size $\epsilon/4$. The perturbation limit $\epsilon = 0.3$ for MNIST, and $\epsilon = 8/255$ for CIFAR-10. For MART, we set $\lambda = 5$. Hyperparameters of the baselines are configured as per their original papers: max margin is set to 0.45 (MNIST) or 12/255 (CIFAR-10) for MMA, maximum criterion value $c_{\max} = 0.5$ for *Dynamic* and $\lambda = 4$ for TRADES.

---

¶For clarity, we set the ticks on x-axis in uniform interval for different $\lambda$.

‖*Standard*, MMA and TRADES have been introduced in Section 3.2. *Dynamic* is the adversarial training with a criterion that dynamically controls the convergence quality of the inner maximization.

Table 2: White-box robustness (accuracy (%) on white-box test attacks) on MNIST and CIFAR-10.

| Defense | MNIST | | | | CIFAR-10 | | | |
|---|---|---|---|---|---|---|---|---|
| | Natural | FGSM | PGD$^{20}$ | CW$_\infty$ | Natural | FGSM | PGD$^{20}$ | CW$_\infty$ |
| *Standard* | 99.11 | 97.17 | 94.62 | 94.25 | 84.44 | 61.89 | 47.55 | 45.98 |
| MMA | 98.92 | 97.25 | 95.25 | 94.77 | **84.76** | 62.08 | 48.33 | 45.77 |
| Dynamic | 98.96 | 97.34 | 95.27 | 94.85 | 83.33 | 62.47 | 49.40 | 46.94 |
| TRADES | **99.25** | 96.67 | 94.58 | 94.03 | 82.90 | 62.82 | 50.25 | 48.29 |
| **MART** | 98.74 | **97.87** | **96.48** | **96.10** | 83.07 | **65.65** | **55.57** | **54.87** |

Table 3: Black-box robustness (accuracy (%) on black-box test attacks) on MNIST and CIFAR-10.

| Defense | MNIST | | | | CIFAR-10 | | | |
|---|---|---|---|---|---|---|---|---|
| | FGSM | PGD$^{10}$ | PGD$^{20}$ | CW$_\infty$ | FGSM | PGD$^{10}$ | PGD$^{20}$ | CW$_\infty$ |
| *Standard* | 96.12 | 95.73 | 95.47 | 96.34 | 79.98 | 80.27 | 80.01 | 80.85 |
| MMA | 96.11 | 95.94 | 95.81 | 96.87 | 80.28 | 80.52 | 80.48 | 81.32 |
| Dynamic | 97.60 | 96.25 | 95.82 | 97.03 | 81.37 | 81.71 | 81.38 | 82.05 |
| TRADES | 97.49 | 96.03 | 95.73 | 97.20 | 81.52 | 81.73 | 81.53 | 82.11 |
| **MART** | **97.77** | **96.96** | **96.97** | **98.36** | **82.75** | **82.93** | **82.70** | **82.95** |

**White-box Robustness.** We evaluate the robustness of all defense models against three types of attacks for both MNIST and CIFAR-10: FGSM, PGD$^{20}$ (20-step PGD with step size $\epsilon/10$), and CW$_\infty$ ($L_\infty$ version of CW optimized by PGD). All attacks have full access to model parameters and are constrained by the same perturbation limit $\epsilon$. The white-box robustness of all defense models are reported in Table 2, where "Natural" denotes the accuracy on natural test images. Our proposed defense MART achieves the best robustness against all three types of attacks on both MNIST and CIFAR-10. Compared with MNIST, the robustness improvements of MART over other baselines are more significant on CIFAR-10. This is because adversarial training on CIFAR-10 is a more challenging problem that may have more misclassified examples during training, and MART can better handle those misclassified examples due to its regularization term in (11). Note that the robustness improvement of MART is not caused by the so-called "obfuscated gradients" (Athalye et al., 2018). This can be verified by two phenomenons: (1) strong test attacks (*e.g.*, PGD$^{20}$) have higher success rates (lower accuracies) than weak test attacks (*e.g.*, FGSM), and (2) white-box test attacks have higher success rates than back-box test attacks (comparing Table 2 with Table 3). Besides, we conduct an additional check using a gradient-free attack SPSA (Uesato et al., 2018). SPSA attack does not obtain lower accuracy than gradient-based attacks like PGD, which confirms that the robustness of MART trained models are not due to gradient masking.

**Black-box Robustness.** Black-box test attacks are crafted from the natural test images by attacking a surrogate model with an architecture that is either a copy of the defense model (MNIST) or a more complex ResNet-50 model (CIFAR-10). Both surrogate models are trained separately from the defense models on the original training sets. The attacking methods used here are: FGSM, PGD$^{10}$, PGD$^{20}$, and CW$_\infty$. The black-box robustness of all defense models are reported in Table 3. Again, the proposed defense MART achieves higher robustness than other baselines. Compared with the white-box results, all defense methods achieve much better robustness against black-box attacks, even close to the natural accuracy. This suggests that adversarial training is indeed a very practical choice for defense scenarios where the target model can be kept secret from potential attackers. It is also observed that robustness on strong attacks like CW$_\infty$ is higher than weak attacks like FGSM, which indicates that strong attacks have less transferability than weak attacks (Madry et al., 2018).

## 4.3 BENCHMARKING THE STATE-OF-THE-ART ROBUSTNESS

In this part, we conduct more experiments on a large-capacity network WideResNet (Zagoruyko & Komodakis, 2016) to benchmark the state-of-the-art robustness, and also explore using unlabeled data for further robustness boost.

**Performance on WideResNet.** We employ WideResNet-34-10 (depth 34 and width 10) to explore the full power of our proposed MART defense method, and also benchmark the state-of-the-art robustness on CIFAR-10. The robustness of all defense models are tested against white-box FGSM, PGD$^{20}$, PGD$^{100}$ and CW$_\infty$ attacks, under the same settings as Section 4.1. We report the robustness

Table 4: White-box robustness (%) on CIFAR-10 using the WideResNet-34-10.

| Defense | Natural | FGSM | | PGD$^{20}$ | | PGD$^{100}$ | | CW$_\infty$ | |
|---|---|---|---|---|---|---|---|---|---|
| | | Best | Last | Best | Last | Best | Last | Best | Last |
| *Standard* | **87.30** | 56.10 | 56.10 | 52.68 | 49.31 | 51.55 | 49.03 | 50.73 | 48.47 |
| Dynamic | 84.51 | 63.53 | 63.53 | 55.03 | 51.70 | 54.12 | 50.07 | 51.34 | 49.27 |
| TRADES | 84.22 | 64.70 | 64.70 | 56.40 | 53.16 | 55.68 | 51.27 | 51.98 | 51.12 |
| MART | 84.17 | **67.51** | **67.51** | **58.56** | **57.39** | **57.88** | **55.04** | **54.58** | **54.53** |

Table 5: White-box robustness (%) on WideResNet with additional unlabeled data.

a ) WideResNet-34-8 with 100K unlabeled data   b ) WideResNet-28-10 with 500K unlabeled data

| Defense | Natural | PGD$^{20}$ | | Defense | Natural | PGD$^{20}$ |
|---|---|---|---|---|---|---|
| UAT++ | 86.04 | 59.41 | | UAT++ | 86.21 | 62.76 |
| RST | 88.24 | 59.60 | | RST | 89.70 | 63.10 |
| MART | 86.68 | 61.88 | | MART | 86.30 | 65.04 |

of both the best and the last epoch models obtained during training in Table 4 $^\|$. For each defense method against each attack, the "best" refers to the highest robustness that ever achieved at different checkpoints. Specifically, against FGSM attack, the best model was found at the last epoch (*e.g.*, "best" is also "last") for all defense methods, while against PGD$^{20}$, PGD$^{100}$ and CW$_\infty$ attacks, the best model was found at the epoch right after the first time learning rate decay (*i.e.*, epoch 76). Our proposed MART outperforms all baseline methods in terms of the robustness of both the best and the last epoch models. Particularly under the most common comparison setting (against PGD$^{20}$ attacks on CIFAR-10), MART improved $\sim 8\%$ over *Standard*, and $\sim 4\%$ even over TRADES for the last epoch model. A similar trend of improvement is also observed for the best epoch model results. Considering the worst case accuracies against all attacks, MART still gains $\sim 6\%$ and $\sim 3.5\%$ robustness improvement over *Standard* and TRADES respectively.

**Boosting with Additional Unlabeled Data.** Here, we evaluate the proposed semi-supervised version of MART and show that it can also benefit from additional unlabeled data and achieves better robustness. Following the exact settings in UAT++ (Uesato et al., 2019) and RST (Carmon et al., 2019), we compare the robustness of MART with them on WideResNet-34-8 and WideResNet-28-10 against PGD$^{20}$ (FGSM$^{20}$) respectively (in the same setting as reported in their paper). The dataset is CIFAR-10 with 100K and 500K unlabeled data extracted from the 80 Million Tiny Images dataset (Torralba et al., 2008). As confirmed in Table 5, our proposed defense MART can also benefit from unlabeled data, and further improves the UAT++ and RST defenses. This again verifies the benefit of differentiating misclassified and correctly classified examples for improving robustness, and further demonstrates the superiority of our proposed method.

## 5   Conclusion and Future Work

In this paper, we investigated the interesting observation that misclassified examples have a recognizable impact on the final robustness of adversarial training, especially for the outer minimization process. Based on this observation, we designed a misclassification aware adversarial risk, which is formulated as adding an misclassification aware regularization to the standard adversarial risk. Following the regularized adversarial risk, we proposed a new defense algorithm, called *Misclassification Aware adveRsarial Training* (MART), with appropriate surrogate loss functions. Experimental results demonstrated that MART can achieve significantly improved adversarial robustness with respect to the state-of-the-art, and can also achieves better robustness with additional unlabeled data.

In the future, we plan to investigate the effect of differentiation of correctly classified/misclassified training examples in the recently proposed certified/provable robustness framework (Cohen et al., 2019; Salman et al., 2019) and explore the potential improvements brought by the differentiation of training examples.

## Acknowledgement

We thank the anonymous reviewers and area chair for their helpful comments. Part of the experiments were done on JD AI Platform "NeuHub" when Yisen Wang was at JD.com.

---

$^\|$Explanations about the "best" and "last" results are in Appendix B.

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

## A  MART ALGORITHM

---

**Algorithm 1** Misclassification Aware adveRsarial Training (MART)

---

1: **Input:** Training data $\{\mathbf{x}_i, y_i\}_{i=1,\dots,n}$, outer iteration number $T_O$, inner iteration number $T_I$, maximum perturbation $\epsilon$, step size for inner optimization $\eta_I$, step size for outer optimization $\eta_O$
2: **Initialization:** Standard random initialization of $h_{\boldsymbol{\theta}}$
3: **for** $t = 1, \dots, T_O$ **do**
4:     Uniformly sample a minibatch of training data $B^{(t)}$
5:     **for** $\mathbf{x}_i \in B^{(t)}$ **do**
6:         $\mathbf{x}_i' = \mathbf{x}_i + \epsilon \cdot \xi$, with $\xi \sim \mathcal{U}(-1, 1)$     # $\mathcal{U}$ is a uniform distribution
7:         **for** $s = 1, \dots, T_I$ **do**
8:             $\mathbf{x}_i' \leftarrow \Pi_{\mathcal{B}_\epsilon(\mathbf{x}_i)}\big(\mathbf{x}_i' + \eta_I \cdot \mathrm{sign}(\nabla_{\mathbf{x}_i'}\mathrm{CE}(\mathbf{p}(\mathbf{x}_i', \boldsymbol{\theta}), y_i))\big)$     # $\Pi(\cdot)$ is the projection operator
9:         **end for**
10:         $\hat{\mathbf{x}}_i' \leftarrow \mathbf{x}_i'$
11:     **end for**
12:     $\boldsymbol{\theta} \leftarrow \boldsymbol{\theta} - \eta_O \sum_{\mathbf{x}_i \in B^{(t)}} \nabla_{\boldsymbol{\theta}}\mathcal{L}(\mathbf{x}_i, y_i, \hat{\mathbf{x}}_i'; \boldsymbol{\theta})$
13: **end for**
14: **Output:** Robust classifier $h_{\boldsymbol{\theta}}$

---

## B  EXPLANATIONS ABOUT THE RESULTS OF TRADES

To make a fair comparison with the latest method TRADES (Zhang et al., 2019), our code is built upon the TRADES framework. We would like to point out that the robustness reported in their paper is the **\*best\*** robustness that occurred during training. To avoid questioning of our results, we have reported both the **\*best\*** and the **\*last\*** results in Table 4, and the **\*best\*** results for TRADES match their original paper.

