# OpenReview forum: "Improving Adversarial Robustness Requires Revisiting Misclassified Examples"
_ICLR.cc/2020/Conference — Accept (Poster)_

### Official Review · AnonReviewer1 · 2019-10-22
**Official Blind Review #1**

**Rating:** 6

**Review:**

Summary:
=======
Neural Networks (NN) have been shown to be susceptible to various adversarial attacks i.e. if we perturb the "x" just a little, the output prediction changes. So, there has been much research devoted to how we can make NNs robust to such attacks. Typically, the adversarial examples that are used to train adversarially robust methods are generated from the correctly classified examples.

This paper first shows empirically that the adversarial examples generated from misclassified examples by the model h_{\theta} are as important as the ones generated from correctly classified examples on the CIFAR dataset. The authors then propose a novel objective for the adversarial risk that incorporates the misclassified examples as a regularizer. Next, a surrogate convex objective is derived which is optimized to come up with a new kind of adversarial training called misclassification aware adversarial training (MART).


Originality:
==========
The proposed objective seems novel to me compared to the existing methods, perhaps less so to someone who is an expert in adversarial methods. That said, I definitely found the addition of a misclassification-based regularization term as intriguing.

However, I was wondering that maybe the previous approaches did not distinguish between correctly vs incorrectly classified examples while generating adversarial examples was because that is a more general setting. This is so because the idea of classification is tied to a certain model h_{\theta} and hence is not model-agnostic by definition. Perhaps the strength of the approach by Madry et. al. 18, which is the closest work to this paper, is in being model-agnostic. I hope I am not missing something!


Quality:
=======
The paper is technically sound and is a solid contribution to the literature on adversarial robustness. The motivation,  experimental results, and ablation studies are all very well executed. The results are shown on MNIST and CIFAR-10 image datasets and it outperforms a host of competitive baseline algorithms.


Clarity:
=======
The paper is well organized and is very well written. The experimental results are very thorough and well explained.


Significance:
============
The paper solves an important problem of dealing with adversarial attacks on deep neural networks. Further, the solution they propose is novel and a seems like a significant improvement over the extend state-of-the-art.


**Experience Assessment:**

I have read many papers in this area.

**Review Assessment: Checking Correctness Of Derivations And Theory:**

I assessed the sensibility of the derivations and theory.

**Review Assessment: Checking Correctness Of Experiments:**

I carefully checked the experiments.

**Review Assessment: Thoroughness In Paper Reading:**

I read the paper at least twice and used my best judgement in assessing the paper.

---

> ### Author Response · Authors · 2019-11-10
> **Response to Reviewer #1**
>
> Thanks for your positive and valuable comments.
>
> Q1: “I was wondering that maybe the previous approaches did not distinguish between correctly vs incorrectly classified examples while generating adversarial examples was because that is a more general setting. This is so because the idea of classification is tied to a certain model h_{\theta} and hence is not model-agnostic by definition. Perhaps the strength of the approach by Madry et. al. 18, which is the closest work to this paper, is in being model-agnostic. I hope I am not missing something!”
>
> A1: We would like to clarify that both our approach and the standard adversarial training by Madry et. al. 18 are of the same nature. Like Madry et. al. 18, our algorithm can also be applied to any models as long as it can output class probability, i.e., $p_k(x_i,\theta)$ in Equation (2). The difference is that our method makes the outer minimization of adversarial training more data-adaptive (with differentiation between misclassified and correctly classified examples).

---

### Official Review · AnonReviewer3 · 2019-10-24
**Official Blind Review #3**

**Rating:** 6

**Review:**

The paper essentially presented a viewpoint, i.e. misclassified examples may have a significant impact on the final robustness.
The authors conducted a series of qualitative experiments to verify
1) Misclassified examples have more impact on the final robustness than correctly classified examples.
2) For misclassified examples, different maximization techniques may have a negligible influence, but minimization techniques play a critical role on the final robustness.
3) The authors proposed a new defense algorithm which focus on generating adversarial examples from misclassified examples during the training. The algorithm was shown to improve the final robustness by revisiting these adversarial misclassified examples.

Generally speaking, the idea, though somewhat straightforward and less elegant, is reasonably presented and also well-motivated. Empirical validations seemed to support the idea and indicated that the proposed approach could improve the adversarial robustness.


There are several major concerns with the paper as follows:

1.	It is good and reasonable to put more emphasis on the misclassification examples, since it is likely that the region of the “perturbation” largely overlaps with the region of the misclassified examples.

2.	However, this may be dealt with in a more elegant or systematic way. In the viewpoint of the reviewer, actually a different emphasis can be imposed on each different data point including both the correctly-classified and mis-classified samples. In another word, a more systematical extension or a metric may actually be developed emphasizing more on mis-classified samples. It is suspected that different samples within the set of mis-classified samples (or even in the set correctly-classified samples) could also have a different influence. It is highly likely that a more elegant and mathematic way can be designed for this purpose.

3.	Further to point 2, in particular, the regulation w.r.t the correctly classified examples may not be ignored. It is interesting to consider differently the correctly-classified examples due to the trade-off between robustness and standard accuracy. This can also be seen in another submission of ICLR2020 (titled Sensible adversarial learning”) which actually discards the mis-classified samples. I would like to see some additional comments, clarification, or discussions on this point.

4.	I am curious to know if outliers would be over-emphasized by the proposed idea. Some discussion or even some illustrations on a synthetic case would be interesting. Will the existence of outliers affect the robustness?

5.	In Table 2, the best and the last results on FGSM are totally identically, I wonder if this is a wrong copy, which should be further clarified and discussed.

6.	 In the Unlabeled Data experiment part, it is better to compare the results with both UAT++ and RST at the same time rather than separately.

7.	Equation (4) shows that the two parts on the right side are added together, is that right? This is inconsistent with Equation (7) and the following description.

8.	In 3.2, the first paragraph, “max-margin adversarial training (MMA) (Ding et al., 2019)” appears to be wrong. The correct reference is “GavinWeiguang Ding, Yash Sharma, Kry Yik Chau Lui, and Ruitong Huang. Max-margin adversarial (mma) training: Direct input space margin maximization through adversarial training. arXiv preprint arXiv:1812.02637, 2018.”.


==================
I have read carefully the response from the authors. The newly added results appear satisfactory to me and I  am generally happy with the clarifications. I have then  adjusted my rating accordingly.

**Experience Assessment:**

I have published in this field for several years.

**Review Assessment: Checking Correctness Of Derivations And Theory:**

I carefully checked the derivations and theory.

**Review Assessment: Checking Correctness Of Experiments:**

I carefully checked the experiments.

**Review Assessment: Thoroughness In Paper Reading:**

I read the paper thoroughly.

---

> ### Author Response · Authors · 2019-11-10
> **Response to Reviewer #3**
>
> Thanks for your valuable feedback. We address them in detail as follows.
>
> Q1 & 2: A suggestion of different emphasis on each different data point including both the correctly-classified and misclassified samples. A more elegant and systematic way can be designed for this purpose.
>
> A1 & 2: Actually, our approach has already imposed a different emphasis on each different data point according to whether it is correctly-classified or misclassified. As presented in Eq. (11), the weight term $1-p$ is applied on both correctly-classified and misclassified examples: examples of high confidence (e.g., correctly classified ones) will have small $1-p$, while low confidence training examples (e.g., misclassified ones) will have large $1-p$, that is, different training examples are treated adaptively (during training) and differently. We agree that there might exist a more principled and elegant way to deal with misclassified examples/correctly classified examples, and we will explore it in our future work.
>
> Q3: Request for additional comments on another submission of ICLR2020 (titled “Sensible adversarial learning”) which considers correctly classified examples differently and discards the misclassified ones.
>
> A3: First of all, we would like to clarify that the ICLR2020 paper “Sensible adversarial learning” is a concurrent submission as our work. After reading the paper, we found that their method explores different treatments on correctly-classified and misclassified examples at the maximization step of adversarial training, while our work considers different treatments at the minimization step of adversarial training, i.e., applying example-wise regularization term on both correctly-classified and misclassified examples based on the weight $1-p$.
>
> Q4: I am curious to know if the existence of outliers affects the robustness?
>
> A4: We would appreciate more information on the type of outliers the reviewer would like to discuss here: 1) outliers that are wrongly labeled, or 2) outliers that are correctly labeled but far away from other data points in the same class?
>
> Q5: In Table 4, the best and the last results on FGSM are identical, I wonder if this is a wrong copy.
>
> A5: This is not a wrong copy. For Table 4, “best” refers to the best checkpoint for each defense under each attack method. For all defense methods against FGSM, the “best” checkpoint just happens to be at the last epoch, which is why the “best” and the “last” results are identical. We have clarified this in the revision.
>
> Q6: In the Unlabeled Data experiment part, it is better to compare the results with both UAT++ and RST at the same time rather than separately.
>
> A6: Thank you for your suggestion. We have updated Table 5, comparing MART with both UAT++ and RST at the same time. Please note that our conclusion remains the same, i.e., the proposed defense MART can also benefit from unlabeled data, and further improves the UAT++ and RST defenses, achieving the state-of-the-art robustness.
>
> Q7: Equation (4) shows that the two parts on the right side are added together, is that right? This is inconsistent with Equation (7) and the following description.
>
> A7: We have added more steps of derivation in Equation (7) to show the consistency between Equation (4) and Equation (7).
>
> Q8: The reference in 3.2, the first paragraph, “max-margin adversarial training (MMA) (Ding et al., 2019)” appears to be wrong.
>
> A8: Thanks for the correction. We have fixed it in the revision.

---

> > ### Comment · AnonReviewer3 · 2019-11-14
> > **About the outliers...**
> >
> > I am happy with the most answers addressed by the authors. For Q4, I think I am mostly concerned about the outliers that are wrongly labeled. Any comments about this?

---

> > > ### Author Response · Authors · 2019-11-14
> > > **The results on outliers**
> > >
> > > Thanks for your clarification. We have run additional experiments with ResNet-18 on CIFAR-10 dataset with manually corrupted wrong labels (e.g., 0% - 20% percent of training examples have labels that have been randomly and uniformly flipped to 9 other incorrect classes). Other experimental settings are the same as Section 4.1. The results are as follows:
> > >
> > > Wrong label rate		        0%		10%		20%
> > > Robustness on PGD-20	55.45%	54.68%  	53.60%
> > >
> > > We can see that our approach is moderately robust to label outliers, i.e., robustness does drop when label outlier rate increases, but only slightly. Note that the state-of-the-art TRADES method on CIFAR-10 (ResNet-18) without any wrong labels is only 50.25% from Table 2.

---

### Official Review · AnonReviewer2 · 2019-10-27
**Official Blind Review #2**

**Rating:** 8

**Review:**

The paper improves adversarial training by introducing two modifications to the loss function: (i) a "boosted" version of the cross-entropy loss that involves a term similar to a large-margin loss, and (ii) weighting the adversarial loss differently depending on how correctly classified an example is. When put together, these modifications achieve state-of-the-art robustness on CIFAR-10, improving over the previously best robust accuracy by about 3.5%. The authors perform multiple ablation studies and demonstrate that their modified loss function also improves when additional unlabeled data is added (again achieving state-of-the-art robustness).

I recommend accepting the paper. The modifications for the loss function are well motivated and improve over the state of the art by a non-trivial amount. Moreover, the authors nicely put their loss function in the context of prior work.

Additional comments:

- In Table 4, are the "best" columns the best checkpoint for the respective column (potentially different checkpoints for different columns) or does "best" refer to a single model (for each row)?

- Is 65.04% (Table 5 b) now the best published robust accuracy on CIFAR-10 (at least to best of the authors' knowledge)? If so, it may be helpful to indicate this to the reader.

- In Figure 2d, it could be insightful to expand the plot further to see the regime where the performance of MART drops substantially.

- In Figure 1, the three plots would be easier to compare if the y-axes were the same.

- From Figure 2, it looks like the gain from the BCE loss is as large as the gain from treating misclassified examples differently. Is this correct?

- I strongly encourage the authors to release their models in a format that is easy for other researchers to use (e.g., PyTorch model checkpoints). This will make it substantially easier for future work to build on the results in this paper.

**Experience Assessment:**

I have published in this field for several years.

**Review Assessment: Checking Correctness Of Derivations And Theory:**

I assessed the sensibility of the derivations and theory.

**Review Assessment: Checking Correctness Of Experiments:**

I assessed the sensibility of the experiments.

**Review Assessment: Thoroughness In Paper Reading:**

I read the paper at least twice and used my best judgement in assessing the paper.

---

> ### Author Response · Authors · 2019-11-10
> **Response to Reviewer #2**
>
> Thanks for your valuable and positive comments. We address them in detail as follows.
>
> Q1: In Table 4, are the "best" columns the best checkpoint for the respective column (potentially different checkpoints for different columns) or does "best" refer to a single model (for each row)?
>
> A1: Here, “best” refers to the best checkpoint for each defense method under each attack method. Empirically, we find that, against FGSM attack,  the “best” was found at the “last” checkpoint, while against other attacks (PGD and CW), the “best” was found right after the first time learning rate decay (i.e., epoch 76). This is consistent for all defense methods.  We have clarified this in the revision.
>
> Q2: Is 65.04% (Table 5 b) now the best published robust accuracy on CIFAR-10 (at least to the best of the authors' knowledge)?
>
> A2: Yes. To the best of our knowledge, this is the best robust accuracy on CIFAR-10 (with the help of 500K unlabeled data) by our submission to ICLR2020. We have indicated this in the revision.
>
> Q3: In Figure 2d, it could be insightful to expand the plot further to see the regime where the performance of MART drops substantially.
>
> A3: We have run additional experiments for larger lambda and updated Figure 2d accordingly. The performance of MART (as well as TRADES) drops gradually as lambda increases from 20 to 50.
>
> Q4: In Figure 1, the three plots would be easier to compare if the y-axes were the same.
>
> A4: Thanks for pointing this out. We have updated Figure 1 to the same y-axes.
>
> Q5: From Figure 2, it looks like the gain from the BCE loss is as large as the gain from treating misclassified examples differently. Is this correct?
>
> A5: Yes, it is correct.
>
> Q6: Encourage to release the models.
>
> A6: Yes, we will release our code and model upon acceptance.

---

### Author Response · Authors · 2020-02-22
**Camera Ready Update**

We have made minor updates to results in Tables 2&3 (the ResNet-18 results). The original results were run on an industrial AI platform with PyTorch 1.0.0 and NVIDIA Tesla V100 GPU, which is no longer accessible since the first author (Y. Wang) has left the company. To better prepare the camera-ready, the first author rerun the experiments on a personal server with PyTorch 1.3.1 and NVIDIA GeForce GTX 2080Ti GPU. The new WideResNet results are within +/- 0.3% of the original results, thus are not updated. The new ResNet-18 results are within +/- 0.8% of the original results, some (e.g., whitebox FGSM, PGD20) are slightly higher than the originally reported results, while others (e.g., whitebox Natural) are slightly lower than the original ones. The detailed new and original results are summarized as follows.


White-box attack	Natural	        FGSM		PGD20	        CW
Original results	83.79		65.34		55.45		54.80
New results		83.07		65.65		55.57		54.87

Black-box attack	FGSM		PGD10	        PGD20	        CW
Original results	83.07		83.52		83.09		83.41
New results		82.75		82.93		82.70		82.95


These updates won’t affect our claims made in the paper.

---

### Decision · Program_Chairs · 2019-12-19

**Decision:**

Accept (Poster)

**Comment:**

This paper presents modifications to the adversarial training loss that yield improvements in adversarial robustness.  While some reviewers were concerned by the lack of mathematical elegance in the proposed method, there is consensus that the proposed method clears a tough bar by increasing SOTA robustness on CIFAR-10.